# ON THE BOTTLENECK OF GRAPH NEURAL NETWORKS AND ITS PRACTICAL IMPLICATIONS

**Uri Alon & Eran Yahav**
Technion, Israel
{urialon,yahave}@cs.technion.ac.il

## ABSTRACT

Since the proposal of the graph neural network (GNN) by Gori et al. (2005) and Scarselli et al. (2008), one of the major problems in training GNNs was their struggle to propagate information between distant nodes in the graph. We propose a new explanation for this problem: GNNs are susceptible to a *bottleneck* when aggregating messages across a long path. This bottleneck causes the *over-squashing* of exponentially growing information into fixed-size vectors. As a result, GNNs fail to propagate messages originating from distant nodes and perform poorly when the prediction task depends on long-range interaction. In this paper, we highlight the inherent problem of over-squashing in GNNs: we demonstrate that the bottleneck hinders popular GNNs from fitting long-range signals in the training data; we further show that GNNs that absorb incoming edges equally, such as GCN and GIN, are *more susceptible* to over-squashing than GAT and GGNN; finally, we show that prior work, which extensively tuned GNN models of long-range problems, suffer from over-squashing, and that breaking the bottleneck improves their state-of-the-art results without any tuning or additional weights. Our code is available at https://github.com/tech-srl/bottleneck/.

## 1 INTRODUCTION

*Graph neural networks* (GNNs) (Gori et al., 2005; Scarselli et al., 2008; Micheli, 2009) have seen sharply growing popularity over the last few years (Duvenaud et al., 2015; Hamilton et al., 2017; Xu et al., 2019). GNNs provide a general framework to model complex structural data containing elements (nodes) with relationships (edges) between them. A variety of real-world domains such as social networks, computer programs, chemical and biological systems can be naturally represented as graphs. Thus, many graph-structured domains are commonly modeled using GNNs.

A GNN layer can be viewed as a message-passing step (Gilmer et al., 2017), where each node updates its state by aggregating messages flowing from its direct neighbors. GNN variants (Li et al., 2016; Veličković et al., 2018; Kipf and Welling, 2017) mostly differ in how each node aggregates the representations of its neighbors with its own representation. However, most problems also require the interaction between nodes that are not directly connected, and they achieve this by stacking multiple GNN layers. Different learning problems require different ranges of interaction between nodes in the graph to be solved. We call this required range of interaction between nodes – the *problem radius*.

In practice, GNNs were observed *not* to benefit from more than few layers. The accepted explanation for this phenomenon is *over-smoothing*: node representations become indistinguishable when the number of layers increases (Wu et al., 2020). Nonetheless, over-smoothing was mostly demonstrated in *short-range* tasks (Li et al., 2018; Klicpera et al., 2018; Chen et al., 2020a; Oono and Suzuki, 2020; Zhao and Akoglu, 2020; Rong et al., 2020; Chen et al., 2020b) – tasks that have small *problem radii*, where a node's correct prediction mostly depends on its local neighborhood. Such tasks include paper subject classification (Sen et al., 2008) and product category classification (Shchur et al., 2018). Since the learning problems depend mostly on short-range information in these datasets, it makes sense why more layers than the problem radius might be extraneous. In contrast, in tasks that also depend on *long-range* information (and thus have larger *problem radii*), we hypothesize that the explanation for limited performance is *over-squashing*. We further discuss the differences between over-squashing and over-smoothing in Section 6.

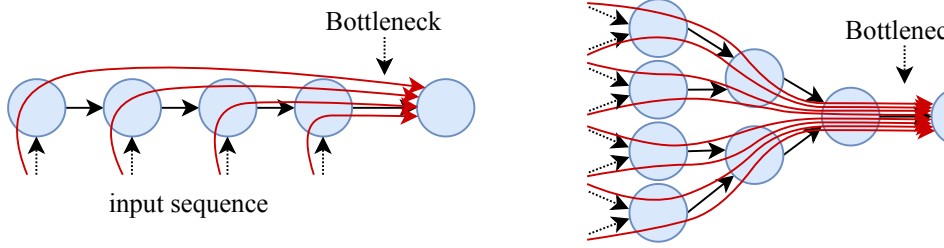

(a) The bottleneck of RNN seq2seq models      (b) The bottleneck of graph neural networks

Figure 1: The bottleneck that existed in RNN seq2seq models (before attention) is strictly more harmful in GNNs: information from a node's exponentially-growing receptive field is compressed into a fixed-size vector. Black arrows are graph edges; red curved arrows illustrate information flow.

To allow a node to receive information from other nodes at a radius of $K$, the GNN needs to have at least $K$ layers, or otherwise, it will suffer from *under-reaching* – these distant nodes will simply not be aware of each other. Clearly, to avoid under-reaching, problems that depend on long-range interaction require as many GNN layers as the range of the interaction. However, as the number of layers increases, the number of nodes in each node's receptive field grows *exponentially*. This causes *over-squashing*: information from the exponentially-growing receptive field is compressed into fixed-length node vectors. Consequently, the graph fails to propagate messages flowing from distant nodes, and learns only short-range signals from the training data.

In fact, the GNN bottleneck is analogous to the bottleneck of sequential RNN models. Traditional seq2seq models (Sutskever et al., 2014; Cho et al., 2014a;b) suffered from a bottleneck at every decoder state – the model had to encapsulate the entire input sequence into a fixed-size vector. In RNNs, the receptive field of a node grows *linearly* with the number of recursive applications. However in GNNs, the bottleneck is asymptotically more harmful, because the receptive field of a node grows *exponentially*. This difference is illustrated in Figure 1.

This work does *not* aim to propose a new GNN variant. Rather, our main contribution is introducing the *over-squashing* phenomenon – a novel explanation for the major and well-known issue of training GNNs for long-range problems, and showing its harmful practical implications. We use a controlled problem to demonstrate how over-squashing prevents GNNs from fitting long-range patterns in the data, and to provide theoretical lower bounds for the required hidden size given the problem radius (Section 5). We show, analytically and empirically, that GCN (Kipf and Welling, 2017) and GIN (Xu et al., 2019) are susceptible to over-squashing *more* than other types of GNNs such as GAT (Veličković et al., 2018) and GGNN (Li et al., 2016). We further show that prior work that extensively tuned GNNs to real-world datasets suffer from over-squashing: breaking the bottleneck using a simple fully adjacent layer reduces the error rate by 42% in the QM9 dataset, by 12% in ENZYMES, by 4.8% in NCI1, and improves accuracy in VARMISUSE, without any additional tuning.

## 2 PRELIMINARIES

A directed graph $\mathcal{G} = (\mathcal{V}, \mathcal{E})$ contains nodes $\mathcal{V}$ and edges $\mathcal{E}$, where $(u, v) \in \mathcal{E}$ denotes an edge from a node $u$ to a node $v$. For brevity, in the following definitions we treat all edges as having the same *type*; in general, every edge can have a type and features (Schlichtkrull et al., 2018).

**Graph neural networks** Graph neural networks operate by propagating neural messages between neighboring nodes. At every propagation step (a graph layer): the network computes each node's sent message; every node aggregates its received messages; and each node updates its representation by combining the aggregated incoming messages with its own previous representation.

Formally, each node is associated with an initial representation $\mathbf{h}_v^{(0)} \in \mathcal{R}^{d_0}$. This representation is usually derived from the node's label or its given features. Then, a GNN layer updates each node's representation given its neighbors, yielding $\mathbf{h}_v^{(1)} \in \mathcal{R}^d$. In general, the $k$-th layer of a GNN is a

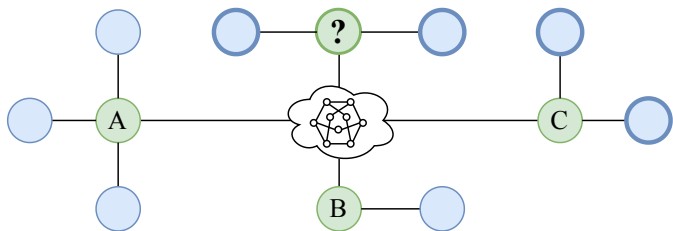

Figure 2: The NEIGHBORSMATCH: green nodes (Ⓐ, Ⓑ, Ⓒ) have blue neighbors (◯) and an alphabetical label. The goal is to predict the label (A, B, or C) of the green node that has the same number of blue neighbors as the target node (?) in the same graph. In this example, the correct label is **C**, because the target node has *two* blue neighbors, like the node marked with C in the same graph.

parametric function $f_k$ that is applied to each node by considering its neighbors:

$$\mathbf{h}_v^{(k)} = f_k \left( \mathbf{h}_v^{(k-1)}, \{\mathbf{h}_u^{(k-1)} \mid u \in \mathcal{N}_v\}; \theta_k \right) \tag{1}$$

where $\mathcal{N}_v$ is the set of nodes that have edges to $v$: $\mathcal{N}_v = \{u \in \mathcal{V} \mid (u, v) \in \mathcal{E}\}$. The total number of layers $K$ is usually determined empirically as a hyperparameter.

The design of the function $f$ is what mostly distinguishes one type of GNN from the other. For example, graph convolutional networks (GCN) define $f$ as:

$$\mathbf{h}_v^{(k)} = \sigma \left( \sum_{u \in \mathcal{N}_v \cup \{v\}} \frac{1}{c_{u,v}} W^{(k)} \mathbf{h}_u^{(k-1)} \right) \tag{2}$$

where $\sigma$ is a nonlinearity such as $ReLU$, and $c_{u,v}$ is a normalization factor often set to $\sqrt{|\mathcal{N}_v| \cdot |\mathcal{N}_u|}$ or $|\mathcal{N}_v|$ (Hamilton et al., 2017). As another example, graph isomorphism networks (GIN) (Xu et al., 2019) update a node's representation using the following definition:

$$\mathbf{h}_v^{(k)} = MLP^{(k)} \left( \left( 1 + \epsilon^{(k)} \right) \mathbf{h}_v^{(k-1)} + \sum_{u \in \mathcal{N}_v} \mathbf{h}_u^{(k-1)} \right) \tag{3}$$

Usually, the last ($K$-th) layer's output is used for prediction: in node-prediction, $\mathbf{h}_v^{(K)}$ is used to predict a label for $v$; in graph-prediction, a permutation-invariant "readout" function aggregates the nodes of the final layer using summation, averaging, or a weighted sum (Li et al., 2016).

## 3 THE GNN BOTTLENECK

Given a graph $\mathcal{G} = (\mathcal{V}, \mathcal{E})$ and a given node $v$, we denote the problem's required range of interaction, the *problem radius*, by $r$. $r$ is generally unknown in advance, and usually approximated empirically by tuning the number of layers $K$. We denote the set of nodes in the receptive field of $v$ by $\mathcal{N}_v^K$, which is defined recursively as $\mathcal{N}_v^1 := \mathcal{N}_v$ and $\mathcal{N}_v^K := \mathcal{N}_v^{K-1} \cup \{w \mid (w, u) \in \mathcal{E} \wedge u \in \mathcal{N}_v^{K-1}\}$.

When a prediction problem relies on long-range interaction between nodes, the GNN must have as many layers $K$ as the estimated range of these interactions, or otherwise, these distant nodes would not be able to interact. It is thus required that $K \geq r$. However, the number of nodes in each node's receptive field grows *exponentially* with the number of layers: $|\mathcal{N}_v^K| = \mathcal{O}(\exp(K))$ (Chen et al., 2018). As a result, an exponentially-growing amount of information is squashed into a fixed-length vector (the vector resulting from the $\sum$ in Equations (2) and (3)), and crucial messages fail to reach their distant destinations. Instead, the model learns only short-ranged signals from the training data and consequently might generalize poorly at test time.

**Example** Consider the NEIGHBORSMATCH problem of Figure 2. Green nodes (Ⓐ, Ⓑ, Ⓒ) have a varying number of blue neighbors (◯) and an alphabetical label. Each example in the dataset is a different graph that has a different mapping from numbers of neighbors to labels. The rest of the graph (marked as 🔮) represents a general, unknown, graph structure. The goal is to predict a label for the target node, which is marked with a question mark ((?)), according to its number of blue

neighbors. The correct answer is **C** in this case, because the target node has *two* blue neighbors, like the node marked with C in the same graph. Every example in the dataset has a different mapping from numbers of neighbors to labels, and thus message propagation and matching between the target node and all the green nodes must be performed *for every graph in the dataset*.

Since the model must propagate information from *all* green nodes before predicting the label, a bottleneck at the target node is inevitable. This bottleneck causes *over-squashing*, which can prevent the model from fitting the training data perfectly. We demonstrate the bottleneck empirically in this problem in Section 4; in Section 5, we provide theoretical lower bounds for the GNN's hidden size. Obviously, adding direct edges between the target node and the green nodes, or making the existing edges bidirectional, could ease information flow for this specific problem. However, in real-life domains (e.g., molecules), we do not know the optimal message propagation structure a priori, and must use the given relations (such as bonds between atoms) as the graph's edges.

Although this is a contrived problem, it resembles real-world problems that are often modeled as graphs. For example, a computer program in a language such as Python may declare multiple variables (i.e., the green nodes in Figure 2) along with their types and values (their numbers of blue neighbors in Figure 2); later in the program, predicting which variable should be used in a specific location (predict the alphabetical label in Figure 2) must use one of the variables that are available in scope based on the required type and the required value at that point. We experiment with this VARMISUSE problem in Section 4.4.

**Short- vs. long-range problems** Much of prior GNN work has focused on problems that were local in nature, with small problem radii, where the underlying inductive bias was that a node's most relevant context is its local neighborhood, and long-range interaction was not necessarily needed. With the growing popularity of GNNs, their adoption expanded to domains that required longer-range information propagation as well, without addressing the inherent bottleneck. In this paper, we focus on problems that *require* long-range information. That is, a correct prediction requires considering the local environment of a node *and* interactions beyond the close neighborhood. For example, a chemical property of a molecule (Ramakrishnan et al., 2014; Gilmer et al., 2017) can depend on the combination of atoms that reside in the molecule's *opposite sides*. Problems of this kind require long-range interaction, and thus, a large number of GNN layers. Since the receptive field of each node grows exponentially with the number of layers, the more layers – over-squashing is more harmful.

In problems that are local in nature (small $r$) – the bottleneck is less troublesome, because a GNN can perform well with only few layers (e.g., $K=2$ layers in Kipf and Welling (2017)), and the receptive field of a node can be exponentially smaller. Domains such as citation networks (Sen et al., 2008), social networks (Leskovec and Mcauley, 2012), and product recommendations (Shchur et al., 2018) usually raise short-range problems and are thus *not* the focus of this paper. So, how long is long-range? We discuss and analyze this question in Section 5.

## 4 EVALUATION

First, we wish to empirically show that the GNN bottleneck exists, and find the smallest values of $r$ that raise over-squashing. We generated a synthetic benchmark that is theoretically solvable; however, in practice, all GNNs fail to reach 100% training accuracy because of the bottleneck (Section 4.1). Second, we examine whether the bottleneck exists in prior work, which addressed real-world problems (Sections 4.2 to 4.4).

### 4.1 SYNTHETIC BENCHMARK: NEIGHBORSMATCH

The NEIGHBORSMATCH problem (Figure 2) is a contrived problem that we designed to provide an intuition to the extent of the effect of over-squashing, while allowing us to control the problem radius $r$, and thus *control the intensity* of over-squashing. We focus on the *training* accuracy of a model, to show that over-squashing prevents models from fitting long-range signals in the training set.

**TREE-NEIGHBORSMATCH** From the perspective of a single node $v$, the rest of the graph may look like a tree of height $K$, rooted at $v$ (Xu et al., 2018; Garg et al., 2020). To simulate this exponentially-growing receptive field, we created an instance of the general NEIGHBORSMATCH problem that we described in Section 3 and portrayed in Figure 2. We instantiated the subgraph in the

middle of the graph (marked as ✿ in Figure 2) as a binary tree of depth $depth$ where the green nodes are its leaves, and the target node is the tree's root. All edges are directed toward the root, such that information is propagated from all nodes toward the target node. The goal, as in Section 3, is to predict a label for the target node, where the correct answer is the label of the green node that has the same number of blue neighbors as the target node. An illustration is shown in Figure 5 in the appendix. This allows us to control the problem radius, i.e., $r = depth$. In this section we observe the bottleneck empirically; in Section 5 we provide a lower bound for the GNN's hidden size given $r$.

**Model** We implemented a network with $r+1$ graph layers to allow an additional nonlinearity after the information from the leaves reaches the target node. Our PyTorch Geometric (Fey and Lenssen, 2019) implementation is available at `https://github.com/tech-srl/bottleneck/`. Our training configuration and hyperparameter ranges are detailed in Appendix A.

**Results** Figure 3 shows the following surprising results: some GNNs fail to fit the dataset starting from $r$=4. For example, the training accuracy of GCN (Kipf and Welling, 2017) at $r$=4 is 70%. At $r$=5, all GNNs fail to perfectly fit the data. Starting from $r$=4, the models suffered from *over-squashing* that resulted in *underfitting*: the bottleneck prevented the models from distinguishing between different training examples, even after they were observed tens of thousands of times. These results clearly show the existence of over-squashing, starting from $r$=4.

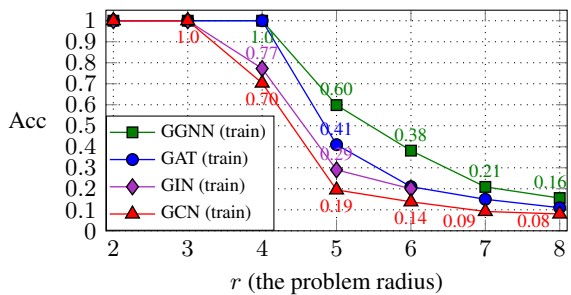

Figure 3: Accuracy across *problem radius* (tree depth) in the NEIGHBORSMATCH problem. Over-squashing starts to affect GCN and GIN even at $r = 4$.

**Why did some GNNs perform better than others?** GCN and GIN managed to perfectly fit $r$=3 at most, while GGNN and GAT also reached 100% accuracy at $r$=4. This difference can be explained by their neighbor aggregation computation: consider the target node that receives messages in the $r$'th step. GCN and GIN aggregate all neighbors *before* combining them with the target node's representation; they thus must compress the information flowing from *all* leaves into a single vector, and *only afterward* interact with the target node's own representation (Equations (2) and (3)). In contrast, GAT uses attention to weight incoming messages given the target's representation: at the last layer only, the target node can ignore the irrelevant incoming edge, and absorb only the relevant incoming edge, which contains information flowing from *half* of the leaves. That is, a single vector compresses *only half* of the information. Since the number of leaves grows exponentially with $r$, it is expected that GNNs that need to compress *only half* of the information (GGNN and GAT) will succeed at an $r$ that is larger by 1. Following Levy et al. (2018), we hypothesize that the GRU cell in GGNNs filters incoming edges as GAT, but perform this filtering as element-wise attention.

**If all GNNs have reached low *training* accuracy, how do GNN-based models usually *do fit* the training data in public datasets of long-range problems?** We hypothesize that they overfit short-range signals and artifacts from the training set, rather than learning the long-range information that was squashed in the bottleneck, and thus generalize poorly at test time.

## 4.2 QUANTUM CHEMISTRY: QM9

We wish to measure over-squashing in existing models. But, how can we measure over-squashing? Instead, we measure whether breaking the bottleneck improves the results of long-range problems.

**Adding a fully-adjacent layer (FA)** In Sections 4.2 to 4.4, we took extensively tuned models from previous work, and modified adjacency in the last layer: given a GNN with $K$ layers, we modified the $K$-th layer to be a *fully-adjacent layer* (FA). A *fully-adjacent layer* is a GNN layer in which every pair of nodes is connected by an edge. In terms of Equations (1) to (3), converting an existing layer to be fully-adjacent means that $\mathcal{N}_v := \mathcal{V}$ for every node $v \in \mathcal{V}$, in that layer only. This does not change the type of layer nor add weights, but only changes adjacency of a data sample in a single layer. Thus, the $K - 1$ graph layers exploit the graph structure using their original sparse topology, and only the

| Property | R-GIN | | R-GAT | | GGNN | |
| --- | --- | --- | --- | --- | --- | --- |
| | base[†] | +FA | base[†] | +FA | base[†] | +FA |
| mu | $2.64\pm0.11$ | $\mathbf{2.54}\pm0.09$ | $\mathbf{2.68}\pm0.06$ | $2.73\pm0.07$ | $3.85\pm0.16$ | $\mathbf{3.53}\pm0.13$ |
| alpha | $4.67\pm0.52$ | $\mathbf{2.28}\pm0.04$ | $4.65\pm0.44$ | $\mathbf{2.32}\pm0.16$ | $5.22\pm0.86$ | $\mathbf{2.72}\pm0.12$ |
| HOMO | $1.42\pm0.01$ | $\mathbf{1.26}\pm0.02$ | $1.48\pm0.03$ | $\mathbf{1.43}\pm0.02$ | $1.67\pm0.07$ | $\mathbf{1.45}\pm0.04$ |
| LUMO | $1.50\pm0.09$ | $\mathbf{1.34}\pm0.04$ | $1.53\pm0.07$ | $\mathbf{1.41}\pm0.03$ | $1.74\pm0.06$ | $\mathbf{1.63}\pm0.06$ |
| gap | $2.27\pm0.09$ | $\mathbf{1.96}\pm0.04$ | $2.31\pm0.06$ | $\mathbf{2.08}\pm0.05$ | $2.60\pm0.06$ | $\mathbf{2.30}\pm0.05$ |
| R2 | $15.63\pm1.40$ | $\mathbf{12.61}\pm0.37$ | $52.39\pm42.5$ | $\mathbf{15.76}\pm1.17$ | $35.94\pm35.7$ | $\mathbf{14.33}\pm0.47$ |
| ZPVE | $12.93\pm1.81$ | $\mathbf{5.03}\pm0.36$ | $14.87\pm2.88$ | $\mathbf{5.98}\pm0.43$ | $17.84\pm3.61$ | $\mathbf{5.24}\pm0.30$ |
| U0 | $5.88\pm1.01$ | $\mathbf{2.21}\pm0.12$ | $7.61\pm0.46$ | $\mathbf{2.19}\pm0.25$ | $8.65\pm2.46$ | $\mathbf{3.35}\pm1.68$ |
| U | $18.71\pm23.36$ | $\mathbf{2.32}\pm0.18$ | $6.86\pm0.53$ | $\mathbf{2.11}\pm0.10$ | $9.24\pm2.26$ | $\mathbf{2.49}\pm0.34$ |
| H | $5.62\pm0.81$ | $\mathbf{2.26}\pm0.19$ | $7.64\pm0.92$ | $\mathbf{2.27}\pm0.29$ | $9.35\pm0.96$ | $\mathbf{2.31}\pm0.15$ |
| G | $5.38\pm0.75$ | $\mathbf{2.04}\pm0.24$ | $6.54\pm0.36$ | $\mathbf{2.07}\pm0.07$ | $7.14\pm1.15$ | $\mathbf{2.17}\pm0.29$ |
| Cv | $3.53\pm0.37$ | $\mathbf{1.86}\pm0.03$ | $4.11\pm0.27$ | $\mathbf{2.03}\pm0.14$ | $8.86\pm9.07$ | $\mathbf{2.25}\pm0.20$ |
| Omega | $1.05\pm0.11$ | $\mathbf{0.80}\pm0.04$ | $1.48\pm0.87$ | $\mathbf{0.73}\pm0.04$ | $1.57\pm0.53$ | $\mathbf{0.87}\pm0.09$ |
| Relative: | | -39.54% | | -44.58% | | -47.42% |

Table 1: Average error rates (5 runs $\pm$ stdev for each property) on the QM9 dataset. The best result for every property in every GNN type is highlighted in bold. Results marked with † were previously reported by Brockschmidt (2020) and reproduced by us.

$K$-th layer is an FA layer that allows the topology-aware node-representations to interact directly and consider nodes beyond their original neighbors. Hopefully, this would ease information flow, prevent over-squashing, and reduce the effect of the previously-existed bottleneck. We re-trained the models using the authors' original code, without performing *any* additional tuning, to rule out hyperparameter tuning as the source of improvement. Statistics of all datasets can be found in Appendix D.

We note that an FA layer is a *simple* solution. Its purpose is merely to demonstrate that over-squashing in GNNs is so prevalent and untreated that *even the simplest solution helps*. Our main contribution is not the solution, but rather, highlighting and explaining the over-squashing *problem*. This simple solution opens the path for a variety of follow-up improvements and solutions for over-squashing.

**Data** The QM9 dataset (Ramakrishnan et al., 2014; Gilmer et al., 2017; Wu et al., 2018) contains ~130,000 graphs with ~18 nodes. Each graph is a molecule where nodes are atoms, and undirected, typed edges are different types of bonds between the atoms. The goal is to regress each graph to 13 real-valued quantum chemical properties such as *dipole moment* and *isotropic polarizability*.

**Models** We modified the implementation of Brockschmidt (2020) who performed an extensive hyperparameter tuning for multiple GNNs, by searching over 500 configurations; we took the same splits and their best-found configurations. For most GNNs, Brockschmidt found that the best results are achieved using $K=8$ layers. This hints that this problem depends on long-range information and relies on both graph structure *and* distant nodes. We re-trained each modified model for each target property using the same code, configuration, and training scheme as Brockschmidt (2020), training each model five times (using different random seeds) for each target property task. We compare the "base" models, reported by Brockschmidt, with our modified and re-trained "+FA" models.

**Results** Results for the top GNNs are shown in Table 1. The main results are that breaking the bottleneck by modifying a single layer to be an FA layer *significantly reduces the error rate*, by 42% on average, across six GNN types. These experiments clearly show evidence for a bottleneck in the original GNN models. Results for the other GNNs are shown in Appendix B due to space limitation.

**Over-squashing or under-reaching?** Barceló et al. (2020) discuss the inability of a GNN node to observe nodes that are farther away than the number of layers $K$. We denote this limitation as *under-reaching*: for every fixed number of layers $K$, local information cannot travel farther than distance $K$ along edges. So, was the improvement of the FA layer in Table 1 achieved thanks to the reduction in over-squashing, or did the FA layer only extend the nodes' reachability and prevent under-reaching? To answer this, we measured the graphs' *diameter* in the QM9 dataset – the maximum shortest path between any two nodes in a graph. We found that the average diameter is $6.35\pm0.91$, the maximum diameter is 10, and the 90'th percentile is 8, while most models were trained with $K=8$ layers. That

|  |  | NCI1 | ENZYMES |
|---|---|---|---|
| No Struct[†] |  | 69.8±2.2 | 65.2±6.4 |
| DiffPool | base[†] | 76.9±1.9 | 59.5±5.6 |
|  | +FA | **77.6**±1.3 | **65.7**±4.8 |
| GraphSAGE | base[†] | 76.0±1.8 | 58.2±6.0 |
|  | +FA | **77.7**±1.8 | **60.8**±4.5 |
| DGCNN | base[†] | 76.4±1.7 | 38.9±5.7 |
|  | +FA | **76.8**±1.5 | **42.8**±5.3 |
| GIN | base[†] | 80.0±1.4 | 59.6±4.5 |
|  | +FA | **81.5**±1.2 | **67.7**±5.3 |

Table 2: Average accuracy (30 runs±stdev) on the biological datasets. † – previously reported by Errica et al. (2020).

|  |  | SeenProj | UnseenProj |
|---|---|---|---|
| GGNN[†] | base[†] | 85.7±0.5 | **79.3**±1.2 |
|  | +FA | **86.3**±0.7 | 79.1±1.1 |
| R-GCN | base[†] | 88.3±0.4 | 82.9±0.8 |
|  | +FA | **88.4**±0.7 | **83.8**±1.0 |
| R-GIN | base[†] | 87.1±0.1 | 81.1±0.9 |
|  | +FA | **87.5**±0.7 | **81.7**±1.2 |
| GNN-MLP | base[†] | 86.9±0.3 | **81.4**±0.7 |
|  | +FA | **87.3**±0.2 | 81.2±0.5 |
| R-GAT | base[†] | 86.9±0.7 | 81.2±0.9 |
|  | +FA | **87.9**±1.0 | **82.0**±1.9 |

Table 3: Average accuracy (5 runs±stdev) on VARMISUSE. † – previously reported by Brockschmidt (2020).

is, at least 90% of the examples in the dataset certainly did *not* suffer from under-reaching, because the number of layers was greater or equal than their diameter. We trained another set of models with 10 layers, which did not show an improvement over the base models. We conclude that the source of improvement was clearly *not* the increased reachability, but instead, the reduction in over-squashing.

**Can larger hidden sizes achieve a similar improvement?** We trained another set of models with *doubled* dimensions. These models achieved only 5.5% improvement over the base model (Appendix B.2), while adding the FA layer achieved 42% improvement using the original dimensions and without adding weights. Consistently, in Section 5 we present an analysis that shows how dimensionality increase is *ineffective* in preventing over-squashing.

**Is the entire FA layer needed?** We experimented with using only a sampled fraction of edges in the FA layer. As Appendix B.3 shows, the fraction of added edges in the last layer correlates with the decrease in error. For example, using only *half* of the possible edges in the last layer (a "semi-adjacent" layer) still reduces the error rate by 31.5% on average compared to "base".

**If all GNNs benefitted from direct interaction between all nodes, maybe the graph structure is not even needed?** We trained another set of models (Appendix B.2) where *all K layers* are FA layers, thus ignoring the original graph topology; these models produced 1500% *higher* (worse) error.

## 4.3 BIOLOGICAL BENCHMARKS

**Data** The NCI1 dataset (Wale et al., 2008) contains 4110 graphs with ~30 nodes on average, and its task is to predict whether a biochemical compound contains anti-lung-cancer activity. ENZYMES (Borgwardt et al., 2005) contains 600 graphs with ~36 nodes on average, and its task is to classify an enzyme to one out of six classes. We used the same 10-folds and split as Errica et al. (2020).

**Models** We used the implementation of Errica et al. (2020) who performed a fair and thorough comparison between GNNs. The final reported result is the average of 30 test runs (10 folds×3 random seeds). Additional training details are provided in Appendix C.

In ENZYMES, Errica et al. found that a baseline that does not use the graph topology *at all* ("*No Struct*") performs better than all GNNs. In NCI1, GIN performed best. We converted the last layer into an FA layer by modifying the implementation of Errica et al., and repeated the same training procedure. We compare the "base" models from Errica et al. with our re-trained "+FA" models.

**Results** Results are shown in Table 2. The main results are as follows: (a) in NCI1, GIN+FA improves by 1.5% over GIN-base, which was previously the best performing model; (b) in ENZYMES, where Errica et al. (2020) found that none of the GNNs exploit the topology of the graph, we find that GIN+FA *does* exploit the structure and improves by 8.1% over GIN-base and by 2.5% over *No Struct*.

On average, models with FA layers relatively reduce the error rate by 12% in ENZYMES and by 4.8% in NCI1. These experiments clearly show evidence for a bottleneck in the original GNN models.

### 4.4 PROGRAMS: VARMISUSE

**Data** VARMISUSE (Allamanis et al., 2018) is a node-prediction problem that depends on long-range information in computer programs. We used the same splits as Allamanis et al. (2018).

**Models** We use the implementation of Brockschmidt (2020) who performed an extensive hyperparameter tuning by searching over 30 configurations for each GNN type. The best results were found using 6-10 layers, which hints that this problem requires long-range information. We modified the last layer to be an FA layer, and used the resulting representations for node classification. We used the same best found configurations as Brockschmidt (2020) add re-trained each model five times.

**Results** Results are shown in Table 3. The main result is that adding an FA layer to all GNNs improves their SeenProjTest accuracy, obtaining a new state-of-the-art of 88.4%. In the *Unseen*ProjTest set, adding an an FA layer improves the results of some of most of the GNNs, obtaining a new state-of-the-art of 83.8%. These improvements are significant, especially since they were achieved on extensively tuned models, without any further tuning by us.

## 5 HOW LONG IS LONG-RANGE?

In this section, we analyze over-squashing combinatorially in the TREE-NEIGHBORSMATCH problem. We provide a combinatorial lower bound for the minimal hidden size that a GNN requires to perfectly fit the data (learn to 100% training accuracy) given its problem radius $r$. We denote the arity of such a tree by $m$ (=2 in our experiments); the counting base as $b$=2; the number of bits in a floating-point variable as $f$=32; and the hidden dimension of the GNN, i.e., the size of a node vector $\mathbf{h}_v^{(k)}$, as $d$.

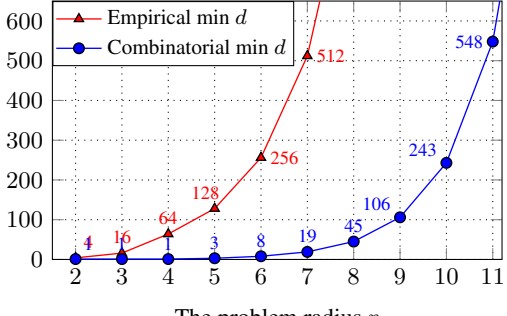

Figure 4: Combinatorial and empirical lower bounds of the model dimension given the problem radius.

A full tree of arity $m$ and problem radius $r$=$depth$ has $m^r$ green label-nodes. All $(m^r)!$ possible permutations of the labels {A, B, C, ...} are valid, disregarding the order of sibling nodes. Thus, the number of label assignments of green nodes is $(m^r)!/(m!)^{m^r-1}$ (there are $m^r - 1$ parent nodes, where the order of each of their $m$ siblings can be permutated). Right before interacting with the target node and predicting the label, a single vector of size $d$ must encapsulate the information flowing from all green nodes (Equations (2) and (3)).[1] Such a vector contains $d$ floating-point elements, each of them is stored as $f$ bits. Overall, the number of possible cases that this vector *can* distinguish between is $b^{f \cdot d}$. The number of possible cases that the vector can distinguish between must be greater than the number of different examples that this vector may encounter in the training data. This requirement is expressed in Equation (4). Considering binary trees ($m$=2), and floating-point values of $f$=32 binary ($b$=2) bits, we get Equation (5):

$$b^{f \cdot d} > \frac{(m^r)!}{(m!)^{m^r-1}} \qquad (4) \qquad\qquad 2^{32 \cdot d} > \frac{(2^r)!}{2^{2^r-1}} \qquad (5)$$

---

[1]The analysis holds for GCN and GIN. Architectures that use the representation of the recipient node to aggregate messages, like GAT, need to compress the information from only *half* of the leaves in a single vector. This increases the final upper bounds on $r$ by up to 1 and demonstrated empirically in Section 4.1.

Since factorial grows faster than an exponent with a constant base, a small increase in $r$ requires a much larger increase in $d$. Specifically, for $d{=}32$ as in the experiments in Section 4.1, the maximal problem radius is as low as $r{=}7$. That is, a model with $d{=}32$ *cannot* obtain 100% accuracy for $r{>}7$.

In practice, the problem is worse; i.e., the empirical minimal $d$ is higher than the combinatorial, because even if a solution to storing some information in a vector of a certain size exists, a gradient descent-based algorithm is not guaranteed to find it. Figure 4 shows the combinatorial lower bound of $d$ given $r$. We also repeated the experiments from Section 4.1 and report the minimal *empirical* $d$ for each value of $r$. As shown in Figure 4, the empirical and the theoretical minimal $d$ grow exponentially with $r$; for example, even $d{=}512$ can empirically fit $r{=}7$ at most.

## 6 RELATED WORK

**Under-reaching** Barceló et al. (2020) found that the expressiveness of GNNs captures only a small fragment of first-order logic. The main limitation arises from the inability of a node to be aware of nodes that are farther away than the number of layers $K$, while the existence of such nodes *can* be easily described using logic. We denote this limitation as *under-reaching*. Nevertheless, even when information is reachable within $K$ edges, we show that this information might be over-squashed along the way. Thus, the *over-squashing* limitation described in this paper is *tighter* than *under-reaching*.

**Over-smoothing** As observed before, node representations become indistinguishable and prediction performance severely degrades as the number of layers increases. The accepted explanation to this phenomenon is *over-smoothing* (Li et al., 2018; Wu et al., 2020; Oono and Suzuki, 2020). This might explain the empirical optimality of few layers in short-range tasks (e.g., only $K{=}2$ layers in Kipf and Welling (2017)). Nonetheless, some problems depend on longer-range information propagation and thus *require* more layers, to avoid *under-reaching*. We hypothesize that in long-range problems, the explanation for the degraded performance is *over-squashing* rather than *over-smoothing*. For further discussion of over-smoothing vs. over-squashing, see Appendix E.

**Avoiding over-squashing** Some previous work avoid over-squashing by various profitable means: Gilmer et al. (2017) add "virtual edges" to shorten long distances; Scarselli et al. (2008) add "supersource nodes"; and Allamanis et al. (2018) designed program analyses that serve as 16 "shortcut" edge types. However, none of these explicitly explained these solutions using over-squashing, and did not identify the bottleneck and its negative cross-domain implications.

## 7 CONCLUSION

We propose a novel explanation to a well known limitation in training graph neural networks: a bottleneck that causes over-squashing. Problems that depend on long-range interaction require as many GNN layers as the desired radius of each node's receptive field. This causes an exponentially-growing amount of information to be squashed into a fixed-length vector. As a result, the GNN fails to propagate long-range information, learns only short-range signals from the training data instead, and performs poorly when the prediction task depends on long-range interaction.

We demonstrate the existence of the bottleneck in a controlled problem, provide theoretical lower bounds for the hidden size given the problem radius, and show that GCN and GIN are more susceptible to over-squashing than GAT and GGNN. We further show that prior models of chemical, biological and programmatical benchmarks suffer from over-squashing by showing that they can be dramatically improved using a simple FA layer. We conclude that over-squashing in GNNs is so prevalent and untreated in some benchmarks – that even the simplest solution helps. Our observations open the path for a variety of follow-up improvements and even better solutions for over-squashing.

### ACKNOWLEDGMENTS

We would like to thank Federico Errica and Marc Brockschmidt for their help in using their frameworks. We are also grateful to (alphabetically): Chen Zarfati, Elad Nachmias, Gail Weiss, Horace He, Jorge Perez, Lotem Fridman, Moritz Plenz, Pavol Bielik, Petar Veličković, Roy Sadaka, Shaked Brody, Yoav Goldberg, and the anonymous reviewers for their useful comments and suggestions.

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

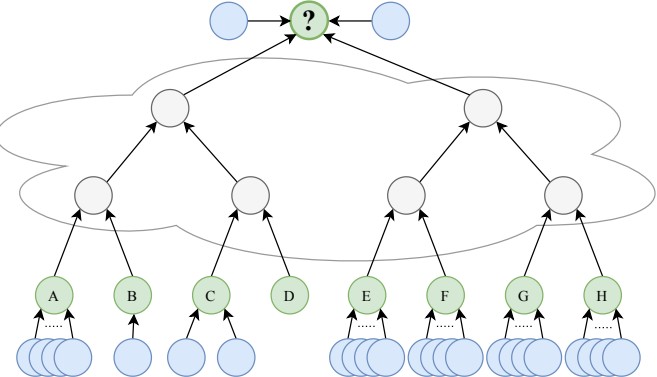

Figure 5: An example of a TREE-NEIGHBORSMATCH, that is an instance of the general NEIGH-BORSMATCH problem that we examine in Section 4. The target node (?) is the root of a tree of $depth$=3 (from the target node to the green nodes). The green nodes (A, B, C, ...) have blue neighbors (◯) and an alphabetical label. The node B has a single blue neighbor; the node C has *two* blue neighbors; and the node D has no blue neighbors; each other green node has another unique number of blue neighbors. The goal it to predict a label for the target node (?) according to its number of blue neighbors. The correct answer is **C** in this example, because the target node has two blue neighbors, like the green node that is marked with C in the same graph. To make a correct prediction, the network must propagate information from *all* leaves toward the target node, and make the decision given a single fixed-sized vector that compresses all this information.

## A    TREE-NEIGHBORSMATCH – TRAINING DETAILS

**Data**    We created a separate dataset for every tree depth (which is equal to $r$, the problem radius) and sampled up to 32,000 examples per dataset. The label of each leaf ("A", "B", "C" in Figure 2) is represented as a one-hot vector. To tease the effect of the bottleneck from the ability of a GNN to count neighbors, we concatenated each leaf node's initial representation with a 1-hot vector representing the number of blue neighbors, instead of creating the blue nodes. The target node is initialized with a learned vector as its (missing) label, concatenated with a 1-hot vector representing its number of blue neighbors. Intermediate nodes are initialized with another learned vector.

**Model**    The network has an initial linear layer, followed by $r + 1$ GNN layers. Afterward, the final target node representation goes through a linear layer and a softmax to predict its label. We experimented with GCN (Kipf and Welling, 2017), GGNN (Li et al., 2016), GIN (Xu et al., 2019) and GAT (Veličković et al., 2018) as the graph layers.

In Section 4.1, we used model dimensions of $d$=32. Larger values led to the exact same trend. We added residual connections, summing every node with its own representation in the previous layer to increase expressivity, and layer normalization which eased convergence. We used the Adam optimizer with a learning rate of $10^{-3}$, decayed by $0.5$ after every 1000 epochs without an increase in training accuracy, and stopped training after 2000 epochs of no training accuracy improvement. This usually led to tens of thousands of training epochs, sometimes reaching 100,000 epochs.

| Property | MLP | | R-GCN | | GNN-FiLM | |
|---|---|---|---|---|---|---|
| | base[†] | +FA | base[†] | +FA | base[†] | +FA |
| mu | 2.36±0.04 | **2.19**±0.04 | 3.21±0.06 | **2.92**±0.07 | 2.38±0.13 | **2.26**±0.06 |
| alpha | 4.27±0.36 | **1.92**±0.06 | 4.22±0.45 | **2.14**±0.08 | 3.75±0.11 | **1.93**±0.08 |
| HOMO | 1.25±0.04 | **1.19**±0.04 | 1.45±0.01 | **1.37**±0.02 | 1.22±0.07 | **1.11**±0.01 |
| LUMO | 1.35±0.04 | **1.20**±0.05 | 1.62±0.04 | **1.41**±0.01 | 1.30±0.05 | **1.21**±0.05 |
| gap | 2.04±0.05 | **1.82**±0.05 | 2.42±0.14 | **2.03**±0.03 | 1.96±0.06 | **1.79**±0.07 |
| R2 | 14.86±1.62 | **12.40**±0.84 | 16.38±0.49 | **13.55**±0.50 | 15.59±1.38 | **11.89**±0.73 |
| ZPVE | 12.00±1.66 | **4.68**±0.29 | 17.40±3.56 | **5.81**±0.61 | 11.00±0.74 | **4.68**±0.49 |
| U0 | 5.55±0.38 | **1.71**±0.13 | 7.82±0.80 | **1.75**±0.18 | 5.43±0.96 | **1.60**±0.12 |
| U | 6.20±0.88 | **1.72**±0.12 | 8.24±1.25 | **1.88**±0.22 | 5.95±0.46 | **1.75**±0.08 |
| H | 5.96±0.45 | **1.70**±0.08 | 9.05±1.21 | **1.85**±0.18 | 5.59±0.57 | **1.93**±0.42 |
| G | 5.09±0.57 | **1.53**±0.15 | 7.00±1.51 | **1.76**±0.15 | 5.17±1.13 | **1.77**±0.05 |
| Cv | 3.38±0.20 | **1.69**±0.08 | 3.93±0.48 | **1.90**±0.07 | 3.46±0.21 | **1.64**±0.10 |
| Omega | 0.84±0.02 | **0.63**±0.04 | 1.02±0.05 | **0.75**±0.04 | 0.98±0.06 | **0.69**±0.05 |
| Relative: | | -40.33% | | -43.40% | | -39.53% |

Table 4: Average error rates and standard deviations on the QM9 targets. Best result for every property in every GNN type is highlighted in bold. Results marked with † were previously reported by Brockschmidt (2020).

To rule out hyperparameter tuning as the source of degraded performance, we experimented with changing activations (ReLu, tanh, MLP, none), using layer normalization and batch normalization, residual connections, various batch sizes, and whether or not the same GNN weights should be "unrolled" over time steps. The presented results were obtained using the configurations that achieved the best results.

**Over-squashing or just long-range?** To rule out the possibility that the long-range itself is preventing the GNNs from fitting the data, we repeated the experiment of Figure 3 for depths 4 to 8, where the distance between the leaves and the target node remained the same, but the amount of over-squashing was as in $r=2$. That is, the graph looks like a tree of $depth=2$, where the root is connected to a "chain" of length of up to 6, and the target node is at the other side of the chain. This setting maintains the long-range as in the original problem, but reduces the amount of information that needs to be squashed. In other words, This setting *disentangles* of the effect of the long-range itself from the effect of the growing amount of information (i.e., from over-squashing). In this setting, *all GNN types managed to easily fit the data to close to 100%* across all distances, showing that the problem is the amount of over-squashing, rather than the long-range itself.

# B  QM9 – ADDITIONAL RESULTS

## B.1  ADDITIONAL GNN TYPES

Because of space limitations, in Section 4.2 we presented results on the QM9 dataset only for R-GIN, R-GAT and GGNN. In this section, we show that additional GNN architectures benefit from breaking the bottleneck using a fully-adjacent layer: GNN-MLP , R-GCN (Schlichtkrull et al., 2018) and GNN-FiLM (Brockschmidt, 2020).

All experiments were performed using the extensively-tuned implementation of Brockschmidt (2020) who experimented with over 500 hyperparameter configurations.

Table 4 contains additional results for GGNN, R-GCN and R-GIN. As shown in Table 4, adding an FA layer significantly improves results across all GNN architectures, for all properties.

## B.2  ALTERNATIVE SOLUTIONS

Table 5 shows additional experiments, all performed using GCN. *base*[†] is the original model of Brockschmidt (2020) as in Table 4. *+FA* is the model that we re-trained with the last layer modified to an FA layer.

| Property | base[†] | +FA | 2×d | All FA | 2×FA | Penultimate FA |
|---|---|---|---|---|---|---|
| mu | 3.21±0.06 | 2.92±0.07 | 2.99±0.08 | 11.52 | 2.89±0.08 | **2.80**±0.08 |
| alpha | 4.22±0.45 | **2.14**±0.08 | 3.57±0.40 | 9.19 | 2.23±0.04 | **2.14**±0.10 |
| HOMO | 1.45±0.01 | 1.37±0.02 | 1.36±1.87 | 9.95 | 1.39±0.02 | **1.34**±0.03 |
| LUMO | 1.62±0.04 | 1.41±0.01 | 1.43±0.04 | 19.13 | 1.42±0.04 | **1.37**±0.02 |
| gap | 2.42±0.14 | 2.03±0.03 | 2.33±0.23 | 24.62 | 2.06±0.05 | **2.00**±0.03 |
| R2 | 16.38±0.49 | 13.55±0.50 | 18.4±0.76 | 168.09 | 13.97±0.56 | **12.92**±0.11 |
| ZPVE | 17.40±3.56 | 5.81±0.61 | 15.8±2.59 | 591.33 | 5.79±0.50 | **4.53**±0.62 |
| U0 | 7.82±0.80 | **1.75**±0.18 | 7.60±2.07 | 188.59 | 1.90±0.1 | 1.98±0.25 |
| U | 8.24±1.25 | 1.88±0.22 | 7.65±1.51 | 189.72 | **1.71**±0.16 | 2.05±0.23 |
| H | 9.05±1.21 | 1.85±0.18 | 8.67±1.10 | 191.11 | 1.83±0.11 | **1.73**±0.14 |
| G | 7.00±1.51 | **1.76**±0.15 | 2.90±1.15 | 173.68 | 1.93±0.11 | 1.96±0.42 |
| Cv | 3.93±0.48 | 1.90±0.07 | 3.99±0.07 | 64.18 | 1.90±0.14 | **1.83**±0.11 |
| Omega | 1.02±0.05 | 0.75±0.04 | 1.03±0.54 | 23.89 | 0.69±0.06 | **0.67**±0.01 |
| relative | 0.0% | -43.40% | -5.50% | +1520% | -43.30% | **-45.2%** |

Table 5: Average error rates and standard deviations on the QM9 targets with GCN using alternative solutions.

$2 \times d$ is a model that was trained with a doubled hidden dimension size, $d = 256$ instead of $d = 128$ as in the base model. As shown, doubling the hidden dimension size leads to a small improvement of only 5.5% reduction in error. In comparison, the +FA model used the original dimension sizes and achieves a much larger improvement of 43.40%.

*All FA* is a model that was trained with *all* GNN layers converted into FA layers, practically ignoring the graph topology. This led to much worse results of more than 1500% higher error. This shows that the graph topology is important in this benchmark, and that a direct interaction between nodes (as in a single FA layer) must be performed in addition to considering the topology.

$2 \times FA$ is a model where the last layer was modified into an FA layer, and an additional FA layer was stacked on top of it. This led to results that are very similar to +FA.

*Penultimate FA* is a model where the FA layer is the penultimate layer (the $K - 1$-th), followed by a standard GNN layer as the $K$-th layer. This led to results that are even slightly better than +FA.

| | base[†] | 0.25× FA | 0.5× FA | 0.75× FA | +FA (as in Table 4) |
|---|---|---|---|---|---|
| Avg. error compared to base[†] | -0% | -8.4% | -31.5% | -37.1% | -43.4% |

Table 6: Average error rates and standard deviations on the QM9 targets with GCN, where we use only a fraction of the edges in the FA layer.

## B.3 PARTIAL-FA LAYERS

We also examined whether instead of adding a "full fully-adjacent layer", we can randomly sample only a fraction of these edges. We randomly sampled only $\{0.25, 0.5, 0.75\}$ of the edges in the full FA layer in every example, and trained the model for each target property 5 times. Table 6 shows the results of these experiments using GCN. *base[†]* is the original model of Brockschmidt (2020) as in Table 4. *+FA* is the model that we re-trained with the last layer modified to an FA layer. $\{0.25, 0.5, 0.75\} \times$ *FA* are the models were only a fraction of the edges in the FA layer were used.

As shown in Table 6, the full FA layer achieves the largest reduction in error (-43.4%), but even adding a fraction of the edges improves the results over the base model. For example, using only *half* of the edges (*0.5× FA*) reduces the error by 31.5%. Overall, the percentage of used edges in the partial-FA layer is correlated with its reduction in error.

## C    BIOLOGICAL BENCHMARKS – TRAINING DETAILS

We used the implementation of Errica et al. (2020) who performed a fair and thorough comparison between GNNs, by splitting each dataset to 10-folds; then, for each GNN type they select a configuration among a grid of 72 configurations according to the validation set; finally, the best configuration for each fold is trained three additional times, early stopped using the validation set, and evaluated on the test set. The final reported result is the average of all 30 test runs (10-folds×3). The final standard deviation is computed among the average results of each of the ten folds.

## D    DATA STATISTICS

### D.1    SYNTHETIC DATASET: TREE-NEIGHBORSMATCH

Statistics of the synthetic TREE-NEIGHBORSMATCH dataset are shown in Table 7.

Table 7: The number of examples, in our experiments and combinatorially, for every value of $depth$.

| $depth$ | # Training examples sampled | Total combinatorial: $\left(2^{depth}!\right) \cdot 2^{depth}$ |
|---|---|---|
| 2 | 96 | 96 |
| 3 | 8000 | $> 3 \cdot 10^5$ |
| 4 | 16,000 | $> 3 \cdot 10^{14}$ |
| 5 | 32,000 | $> 10^{36}$ |
| 6 | 32,000 | $> 10^{90}$ |
| 7 | 32,000 | $> 10^{217}$ |
| 8 | 32,000 | $> 10^{509}$ |

### D.2    QUANTUM CHEMISTRY: QM9

Statistics of the quantum chemistry QM9 dataset, as used in Brockschmidt (2020) are shown in Table 8.

Table 8: Statistics of the QM9 chemical dataset (Ramakrishnan et al., 2014) as used by Brockschmidt (2020).

|  | Training | Validation | Test |
|---|---|---|---|
| # examples | 110,462 | 10,000 | 10,000 |
| # nodes - average | 18.03 | 18.06 | 18.09 |
| # nodes - standard deviation | 2.9 | 2.9 | 2.9 |
| # edges - average | 18.65 | 18.67 | 18.72 |
| # edges - standard deviation | 3.1 | 3.1 | 3.1 |

### D.3    BIOLOGICAL BENCHMARKS

Statistics of the biological datasets, as used in Errica et al. (2020), are shown in Table 9.

### D.4    VARMISUSE

Statistics of the VARMISUSE dataset, as used in Allamanis et al. (2018) and Brockschmidt (2020), are shown in Table 10.

Table 9: Statistics of the biological datasets, as used by Errica et al. (2020).

|  | NCI1 (Wale et al., 2008) | ENZYMES (Borgwardt et al., 2005) |
| --- | --- | --- |
| # examples | 4110 | 600 |
| # classes | 2 | 6 |
| # nodes - average | 29.87 | 32.63 |
| # nodes - standard deviation | 13.6 | 15.3 |
| # edges - average | 32.30 | 64.14 |
| # edges - standard deviation | 14.9 | 25.5 |
| # node labels | 37 | 3 |

Table 10: Statistics of the VARMISUSE dataset (Allamanis et al., 2018) as used by Brockschmidt (2020).

|  | Training | Validation | UnseenProject Test | SeenProject Test |
| --- | --- | --- | --- | --- |
| # graphs | 254360 | 42654 | 117036 | 59974 |
| # nodes - average | 2377 | 1742 | 1959 | 3986 |
| # edges - average | 7298 | 7851 | 5882 | 12925 |

## E    DISCUSSION: OVER-SMOOTHING VS. OVER-SQUASHING

Although *over-smoothing* and *over-squashing* are related, they are disparate phenomena that occur in different types of problems. For example, imagine a triangular graph containing only three nodes, where every node has a scalar value, an edge to each of the other nodes, and needs to compute a function of its own value and the other nodes' values. The problem radius $r$ in this case is $r=1$. As we increase the number of layers, the representations of the nodes might become indistinguishable, and thus suffer from *over-smoothing*. However, there will be *no over-squashing* in this case, because there is no growing amount of information that is squashed into fixed-sized vectors while passing long-range messages. Contrarily, in the TREE-NEIGHBORSMATCH problem, there is no reason for over-smoothing to occur, because there are no two nodes that can converge to the same representation. A node in a "higher" level in the tree contains twice the information than a node in a "lower" level. Thus, this is a case where *over-squashing can occur without over-smoothing*.

