# OpenReview forum: "On the Bottleneck of Graph Neural Networks and its Practical Implications"
_ICLR.cc/2021/Conference — ICLR 2021 Poster_

### Official Review · AnonReviewer1 · 2020-10-24
**simple solution to the bottleneck problem but more ablation studies or model variation could be added**

**Rating:** 6
**Confidence:** 4

**Review:**

The paper analyzes the problem bottleneck in graph neural network message passing, points out that the over-squashing issue results in poor performance in prediction tasks that depend on long-range information on the graph, proposes a simple solution that removes the topology in the last layer, experiments on 3 different datasets, shows improvement over baselines and provides combinatorial analysis.

Strength:
1. The paper provides intuitive examples why long range information is needed in some tasks and why the message passing bottleneck is happening
2. The proposed method (FA) is tested on 3 datasets with multiple bench marks. The results look promising.
3. The combinatorial analysis provides insights on how the bottleneck is limiting GNN and the max depth that a GNN can perfectly fit.

Weakness and suggestions:
1. The paper brings up short vs long-range problems. However, there's no clear definition of what is a short vs. long-range problem. Can you provide a more formal definition of long range interaction on graph? How long is long range? Is there any quantitative measurement to decide that? Is this long range problem task-specific? data-specific? or a combination?
2. In the experiments, the improvement for VarMisuse does not seem as impressive as the other two datasets. Can you provide some insights what is causing this difference in improvement?
3. While the paper proposes a simple solution (FA) to the bottleneck problem, there's not much detail for readers to understand empirically or analytically the extent to which bottleneck issue is mitigated. The experiments do provide some justification and proof that the method would work, but maybe a way to directly measure that indeed more information is passed through is even better.
4. I'm expecting to see more ablation studies or variations of the proposed method, e.g. how would the model perform if applying FA in layers other than the last? instead of a fully-adjacent layer, what about make 50%, or 75% of the nodes directly connected to the node beyond the original neighbor?

---

> ### Author Response · Authors · 2020-11-12
> **Response to AnonReviewer1**
>
> Thank you for taking the time to review our paper and for your kind words!
> Please see our detailed response below.
>
>
> > How long is long range? Is there any quantitative measurement to decide that? Is this long range problem task-specific? data-specific? or a combination?
>
> The distinction between short- and long-range problems is problem-specific and data-specific.
> In most GNN problems and benchmarks, we can only guess the type of problem using our domain knowledge.
> If we could measure, in advance, the needed range of interaction in the graph, we could have decided on the number of needed GNN layers in advance. In practice, the number of GNN layers is determined empirically. To the best of our knowledge, there is no systematic way of determining the number of needed layers, and thus not the "effective range".
>
> Intuitively, short-range tasks are tasks where the classification of a node depends mostly on the information in its local surrounding environment. Such tasks include paper subject classification because a paper's subject is likely to be similar to the subject of the papers it cites.
> Long-range tasks are tasks where the classification of a node can depend on arbitrarily distant nodes. For example, in a computer program, predicting the value in a certain location in the code can depend on a definition many lines earlier.
>
> For many long-range tasks, the whole point is the fact that the exact required length is not known apriori.
> In Section 4.1, we show that in the worst case, GNNs fail to pass long-range messages starting from a distance as low as 4.
>
> > Why is the improvement in VarMisuse more minor than the other datasets?
>
> The VarMisuse benchmark is unique in the sense that it includes and relies on handcrafted "expert edges" that were produced by program analysis. The benchmark included several types of these edges, like "ComputedFrom" and "GuardedByNegation", which serve as "shortcuts" between distant nodes that have a significant effect on each other. These shortcuts relieve some of the over-squashing pressure by passing messages between "important" distance nodes directly.
>
> The designers of this dataset (Allamanis et al., ICLR'2018) used their domain expertise to guide the GNN in interacting directly and passing messages between specific distant nodes.
> However, in most domains (such as molecules), we do not have the expert knowledge to tell which distant nodes are "important" for each other, and the common practice is to feed the molecule's structure as the graph.
>
> Although this is an extensively-tuned benchmark, both in hyperparameters and in the engineering of shortcut edges, adding the FA layer still improves the state-of-the-art-results in the "SeenProj" and "UnseenProj" test sets, in almost all GNN types.
>
>
>
> Here are the updated results for VarMisuse, which we will update in the paper:
>
> |                 |        | GGNN            | R-GCN          | R-GIN          | GNN-MLP        | R-GAT          |
> |-----------------|-----------|-----------------|-----------------|-----------------|-----------------|-----------------|
> | SeenProj Test   | base | 85.7$\pm$0.5 | 88.3$\pm$0.4 | 87.1$\pm$0.1 | 86.9$\pm$0.3 | 86.9$\pm$0.7 |
> |                 | +FA | **86.3**$\pm$0.7 | **88.4**$\pm$0.7 | **87.5**$\pm$0.7 | **87.3**$\pm$0.2 | **87.9**$\pm$1.0 |
> |-------------------------|-----------|-----------------|-----------------|-----------------|-----------------|-----------------|
> | UnseenProj Test | base | **79.3**$\pm$1.2 | 82.9$\pm$0.8 | 81.1$\pm$0.9 | **81.4**$\pm$0.7 | 81.2$\pm$0.9 |
> |                 | +FA | 79.1$\pm$1.1 | **83.8**$\pm$1.0 | **81.7**$\pm$1.2 | 81.2$\pm$0.5 | **82.0**$\pm$1.9 |
>
> All GNNs benefit from the FA layer in the "seen projects, most GNNs benefit from the FA layer in the "unseen projects", and we obtain a new SOTA accuracy of **83.8%** on the "UnseenProj Test" set.

---

> > ### Author Response · Authors · 2020-11-12
> > **Response (2/2)**
> >
> > >The paper proposes a simple solution (FA) to the bottleneck problem... The experiments do provide some justification and proof that the method would work, is there a way to directly measure that indeed more information is passed through?
> >
> > We agree that directly measuring the passed information would be beneficial, but unfortunately, we are not aware of a way to measure the passed information in the general case.
> > We designed the synthetic NeighborsMatch problem to give intuition to the extent of the effects of over-squashing, and to allow us to *control the intensity of over-squashing*.
> >
> > In public datasets, it is difficult (or impossible?) to measure the passed information.
> > We introduce the fully-adjacent (FA) layer to demonstrate that over-squashing in GNNs is *so prevalent and untreated* that *even the simplest solution helps*.
> > Thus, showing that this simplest solution *improves the results on extensively-tuned models* is evidence that the original models are affected by over-squashing.
> >
> > The important point here is the evidence that the **problem exists** in existing state-of-the-art models, rather than the simple solution itself. This simple solution opens the path for a variety of exciting follow-up improvements and even better solutions for over-squashing.
> >
> > >I'm expecting to see more ablation studies or variations of the proposed method
> >
> > We performed more ablations in Appendix B. We show that:
> > 1. Doubling the hidden size of the GNN helps, but only in reducing the average error by about 5% (compared to 43% of adding the FA layer)
> > 2. Making *all layers* FA (i.e., ignoring the graph structure) performs much worse.
> > 3. Adding *two* final FA layers performs similarly to a single FA layer.
> >
> > >how would the model perform if applying FA in layers other than the last?
> >
> > We performed another set of experiments when making the *penultimate* layer an FA layer (trained 5 times for each target property), here are the results:
> >
> > | Model | Task 1 | &nbsp;&nbsp;&nbsp;&nbsp;&nbsp;&nbsp;2  |&nbsp;&nbsp;&nbsp;&nbsp;&nbsp;&nbsp; 3|&nbsp;&nbsp;&nbsp;&nbsp;&nbsp;&nbsp; 4 |&nbsp;&nbsp;&nbsp;&nbsp;&nbsp;&nbsp;  5|&nbsp;&nbsp;&nbsp;&nbsp;&nbsp;&nbsp; 6 | &nbsp;&nbsp;&nbsp;&nbsp;&nbsp;&nbsp; 7| &nbsp;&nbsp;&nbsp;&nbsp;&nbsp;&nbsp; 8| &nbsp;&nbsp;&nbsp;&nbsp;&nbsp;&nbsp; 9| &nbsp;&nbsp;&nbsp;&nbsp;&nbsp;&nbsp; 10| &nbsp;&nbsp;&nbsp;&nbsp;&nbsp;&nbsp; 11| &nbsp;&nbsp;&nbsp;&nbsp;&nbsp;&nbsp; 12|&nbsp;&nbsp;&nbsp;&nbsp;&nbsp;&nbsp; 13 | &nbsp; Rel. to base:|
> > |---------------|-----------:|-----------------:|-----------------:|-----------------:|-----------------:|-----------------:|-----------:|-----------:|-----------:|-----------:|-----------:|-----------:|-----------:|------------------:|
> > |base (Brockschmidt, ICML'2020) | 3.21 | 4.22 | 1.45 | 1.62 | 2.42 | 16.4 | 17.4 | 7.82 | 8.24 | 9.05 | 7.00 | 3.93 | 1.02 |          |
> > |+FA (as in the paper) | 2.92 | **2.14** | 1.37 | 1.41 | 2.03 | 13.6 | 5.81 | **1.75** | 1.88 | 1.85 | **1.76** | 1.90 | 0.75 | **-43.4** |
> > |FA as the penultimate layer | **2.80** | **2.14** | **1.34** | **1.37** | **2.00** | **12.9** | **4.53** | 1.98 | 2.05 | **1.73** | 1.96 | **1.83** | **0.67** | **-45.2** |
> >
> >
> >
> > Using an FA layer as the penultimate layer (followed by a standard GNN layer) performs even slightly better than a final FA layer. We will include these results in the next revision.

---

> > > ### Author Response · Authors · 2020-11-13
> > > **Additional Results for Reviewer1's question**
> > >
> > > >what about 50%, or 75% fully-connected layer?
> > >
> > > We performed the following experiment - instead of adding a full fully-adjacent layer, we randomly sampled only {0.25, 0.5, 0.75} of the edges in the full FA layer for every example, and trained the model for each of the 13 target properties 5 times.
> > >
> > > |                             	| &nbsp;&nbsp;&nbsp; base (Brockschmidt, ICML'2020) 	| &nbsp;&nbsp;&nbsp; 0.25 FA 	| &nbsp;&nbsp;&nbsp; 0.5 FA 	| &nbsp;&nbsp;&nbsp; 0.75 FA 	| &nbsp;&nbsp;&nbsp; +FA (as in the paper) 	|
> > > |-----------------------------	|--------------------------------:	|---------:	|--------:	|---------:	|-----------------------:	|
> > > | Avg. error compared to base 	| 0  | -8.4% | -31.5% |  -37.1% | -43.4% |
> > >
> > >
> > > These results show that the full FA layer achieves the largest reduction in error, but even a fraction of the edges allows partially relieving over-squashing and improves the results over the base model. Overall, the percentage of used edges in the partial-FA layer is correlated with its reduction in error.
> > > These results open the path for a variety of new research directions, such as how to *choose* the edges in a more clever way.

---

### Official Review · AnonReviewer2 · 2020-10-28
**This paper introduces a new bottleneck of GNNs: the over squashing. It analyzes this problem in detail with supporting experiments and results. This bottleneck is quite worthy of thinking when designing deep GNNs.**

**Rating:** 5
**Confidence:** 4

**Review:**

This paper introduces the over-squashing problem in GNNs. The problem is more evident in the graph-structured data where both local neighbors and long-range interactions are required. The paper designs experiments to show that the existing extensively-turned, GNN-based models suffer from the over-squashing problem in long-range prediction tasks. It also shows the improvement of breaking this bottleneck by adding a fully-adjacent layer.

Developing deep GNNs is a popular topic and attracts significant efforts. However, training a deep GNN may face with several problems. Besides the over-smoothing, this paper introduces a new type of bottlenecks: over-squashing. It happens when the GNN needs to pass the information of interactions from long-range neighbors. In this case, exponentially increased information is squashed into a fixed-length vector. I personally believe this bottleneck is quite crucial when we develop deep GNNs.

The overall quality and clarity of this paper are good. It clearly and detailedly explains the over-squashing in long-range problems, and distinguishes it from other similar bottlenecks. It also tries to show the existence of the over-squashing in four different types of datasets. The training details and results are also included. Meanwhile, to my best knowledge, this is the first paper that highlights and analyzes the over-squashing problem in GNNs in detail, and I believe this bottleneck is of vital importance when we design deep GNNs. Therefore, I think the originality and significance of this work are also quite high.

This paper proposes a solution to the over-squashing, which is to add a fully-adjacent layer to the extensively-tuned state-of-the-art model. By comparing the performance of the modified models and the state-of-the-art models in the four datasets mentioned above, we can clearly see an improvement in terms of the accuracy and the error rates. However, although the authors say that this solution is very simple and mainly for the demonstration purpose, I believe there may be a potential problem within this solution. When we add the fully adjacent layer to the original model, we actually change the data directly. That means the modified model and the original model are trained on different datasets. In this case, I am concerned that we cannot compare these two models directly.  Although the authors clarify that the improvement is not from the under-reaching by considering the graph’s diameter in the QM9 dataset, I am still not fully convinced that we can safely rule out the possibility that the improvement is simply because the changed data makes the tasks easier for GNNs. Since the paper shows the existence of the over-squashing by comparing the performance of these two types of models, I believe it would be much better and more convincing if it can propose a new solution that can avoid changing the data directly.


Overall, this paper has the following pros and cons. I believe this paper is of high quality and clarity with supporting experiments and analysis. The bottleneck of over-squashing is also a new problem and worth considering when we develop deep GNNs. But the solution proposed by the authors may be potentially problematic and hence, weaken the results and the conclusions.

---

> ### Author Response · Authors · 2020-11-12
> **Response to AnonReviewer2**
>
> Thank you for taking the time to review our paper and for your kind words!
> We were pleased to read that you "*personally believe this bottleneck is quite crucial when we develop deep GNNs*", that you "*believe this bottleneck is of vital importance when we design deep GNNs. Therefore, I think the originality and significance of this work are also quite high*".
>
>
> Please see our detailed response below.
>
> > when we add the fully adjacent layer to the original model, we actually change the data directly. That means the modified model and the original model are trained on different datasets.
>
> We do **not** believe that these are "different datasets", because adding the FA layer is a *systematic* modification of the network (not a manual selection, not preferring specific examples, nor adding new examples). The examples are exactly the same in the original and the FA model, and the modification is the same across all examples and datasets.
>
> Also note, that if we call this "trained on different datasets", are LSTMs and Transformers trained for a seq2seq task on the same dataset - also incomparable, because their connectivity is different?
>
> We introduce the fully-adjacent (FA) layer to demonstrate that over-squashing in GNNs is *so prevalent and untreated* that *even the simplest solution helps*.
> Thus, showing that this simplest solution *improves the results on extensively-tuned models* is evidence that the original models suffer from over-squashing.
>
> As you write: "*The bottleneck of over-squashing is  a new problem and worth considering when we develop deep GNNs*"
> and "*this is the first paper that highlights and analyzes the over-squashing problem in GNNs in detail*" - these are the important points here, the evidence that the **bottleneck exists** in state-of-the-art models, rather than the simple solution itself.
> This simple solution opens the path for a variety of follow-up improvements and even better solutions for over-squashing.

---

### Official Review · AnonReviewer3 · 2020-10-29
**R3**

**Rating:** 5
**Confidence:** 4

**Review:**

In this paper, the authors identify the 'over-squashing' issue of GNNs. Specifically, the contributions include: 1) highlighting this ‘over-squashing’ bottleneck that affects the representation ability of GNNs; 2) Experiments on both synthetic data and real-world to validate their hypothesis. In general, the paper is structured well and easy to follow. However, there’re a few questions the authors need to address:

First, the differences between over-smoothing and over-squashing has not been discussed thoroughly. Though the authors provide a paragraph in Sec. 6 to bring up the similarity between these two problems, it is not clear what are the actual differences between them. I suggest the authors to provide more explanation or experimental results to validate they are indeed different problems. In addition, the authors mention ‘bottlenecks’ in a few places of this paper but do not provide an explicit definition of this 'bottleneck`' thus it is unclear to readers what it indicates.

Also, I suggest the authors provide more rationale on why making the graph to be fully connected in the last layer of GNN can break the bottleneck introduced by this over-squashing issue. It is interesting to see multiple GNNs perform better by introducing this change, but it would be better if the authors can elaborate on the insights behind adopting this technique.

It is good to see the authors attempt to validate their hypothesis by first using a synthetic dataset, I suggest the authors to further justify why it is the over-squashing issue that makes the GNNs fail to perform well when the number of layer increases. There can be multiple reasons that GNNs fail in this case, e.g., over-smoothing, and analyzing why these models fail helps readers better understand the motivation of conducting this synthetic data evaluation.

For the other experiments, one question is that why the authors use a different set of GNN implementations in the biological benchmarks (Sec. 4.3) rather than the implementation of Brockschmidt used in Sec. 4.2 and 4.4. I suggest the authors further justify the models used here.

I also suggest the authors to revise the caption of Fig. 2 as it is unclear what ‘predict the label C’ means without checking the content in Sec. 3.

In summary, though the papers provide some interesting insights in introducing a fully-connected graph structure in the last layer of GNN. There are still issues the authors need to address.

---

> ### Author Response · Authors · 2020-11-12
> **Response to AnonReviewer3**
>
> Thank you for your detailed review!
> You raise important points that we think are addressable within the discussion phase.
> Please see our detailed response below.
>
> > What is the differences between over-smoothing and over-squashing?
>
> We believe that although related, over-smoothing and over-squashing are different problems, and there are cases where one occurs without the other.
>
> Over-smoothing occurs when node representations become indistinguishable as the number of layers increases. For example, imagine a triangular graph of only three nodes: A$\leftrightarrow$B$\leftrightarrow$C$\leftrightarrow$A .
> If we used too many GNN layers, the representations of A, B and C might become indistinguishable, and thus suffer from **over-smoothing**. However, there will be **no over-squashing** in this case, because the amount of information that needs to be encoded does not grow and is not beyond the capacity of the node representations. There is no growing amount of information that is squashed into fixed-sized vectors while passing long-range messages.
>
> In the other direction - in the case of the synthetic TreeNeighborsMatch problem, there is no reason for over-smoothing to occur, because there are no two nodes that can converge to the same representation, because a node in a "higher" level (in Figure 5) contains twice the information than the node in the "lower" level. Further, the edges are directed and point toward the root. Thus, this is a case where we have **over-squashing without over-smoothing**.
>
> We will elaborate on this in the discussion in Section 6.
>
> >There can be multiple reasons that GNNs fail in the case of the synthetic dataset, e.g., over-smoothing
>
> We argue that this is very unlikely, as detailed above.
>
> To rule out additional reasons, we experimented with changing activations (ReLu, tanh, MLP, none), layer/batch norm, residual connections, batch size, and whether or not the same GNN weights should be "unrolled" over time steps. The presented results were obtained using the configurations that achieved the best results, and our code is supplied in the supplementary material.
>
> >why making the graph to be fully connected in the last layer of GNN can break the bottleneck introduced by over-squashing?
>
> Because adding an FA layer allows both learning the graph topology using the first K-1 layers, and then sending these topology-aware representations to long-range nodes directly.
> Adding an FA layer relieves the first K-1 layers from making an effort to pass long-range messages. The model will not have to optimize these K-1 layers for the objective of compressing long-range messages, because this will be achieved by the last FA layer, which allows a direct passing of long-range messages.
> Thus, useful information can be obtained directly from the original source node, without being squashed.
>
>
> Note that we introduce the fully-adjacent (FA) layer only to demonstrate that over-squashing in GNNs is *so prevalent and untreated* that *even the simplest solution helps*.
> Thus, showing that this simplest solution *improves the results on extensively-tuned models* is evidence that the original models suffer from over-squashing.
> The important point here is the evidence that the **bottleneck exists** in state-of-the-art models, rather than the simple solution itself.
>
>
> > why the authors use a different set of GNN implementations in the biological benchmarks (Sec. 4.3) rather than the implementation of Brockschmidt used in Sec. 4.2 and 4?
>
> We experiment with different implementations to rule out the possibility that the problem is in the original implementations themselves.
>
> We could not use the biological benchmarks using the implementation of Brockschmidt, because the implementation of Brockschmidt was not designed for the biological benchmarks, and Brockschmidt did not perform hyperparameter tuning for these datasets.
>
> We deliberately used GNNs that were implemented and extensively tuned by *others*, and used the same benchmarks that these implementations were designed and tuned for.
> Neither Brockschmidt (ICML'2020) nor Errica et al. (ICLR'2020) are authors of our paper.
>
> >I also suggest the authors to revise the caption of Fig. 2
>
> Thank you, we will improve this in the next revision.

---

### Official Review · AnonReviewer4 · 2020-10-31
**An important observation for GNN limitation with detailed analysis and a simple solution with impressive results**

**Rating:** 8
**Confidence:** 5

**Review:**

This paper identifies the inherent problem of over-squashing that exists in popular information propagation mechanism in GNN. The hidden dimension of a node vector is fixed while the amount of information need to be preserved can grow exponentially (vs linearly in RNN decoder) with the increase of the depth of the networks. This problem becomes critical when modeling long range interaction is required. Then, the paper provides a simple and intuitive solution which is shown to be effective on both synthetic and multiple real-world datasets.

Strength:
- The over-squashing problem is important and inherent in (most of if not all) existing information propagation mechanisms in GNN. Considering the prevalence of GNN, clearly identifying and analyzing the problem can significantly contribute to the community and potentially open a new direction of research.
- The paper provides theoretically analysis on the bottleneck, and further analyzes the relationship of required depth and the hidden dimension, showing the simply increasing the hidden dimension will not solve the problem.
- The paper provides a simple yet effective solution to mitigate the problem, supported by extensive empirical results on multiple datasets.

Minor comments:
- Though the baseline methods are relatively well-known, it is will be still be easier to follow if citations are provided when baseline approaches first appear, e.g., mentioning GGNN (Li et al 2016) and GAT (Velickovic et al 2018) at Page 4. These information is not available until Page 11.
- In Section 4.1, in GAT soft-attention is used, thus the model adaptively combines information from adjacent edges, rather than only consider half of them (which may require hard-attention).

---

> ### Author Response · Authors · 2020-11-12
> **Response to AnonReviewer4**
>
> Thank you for your detailed review and for your kind words!
>
> We were pleased to read that "*The over-squashing problem is important and inherent*", and that "*clearly identifying and analyzing the problem can significantly contribute to the community and potentially open a new direction of research*". These are indeed our main messages.
>
> >This paper identifies the inherent problem of over-squashing that exists in popular information propagation mechanism in GNN. The hidden dimension of a node vector is fixed while the amount of information need to be preserved can grow exponentially (vs linearly in RNN decoder) with the increase of the depth of the networks. This problem becomes critical when modeling long range interaction is required. Then, the paper provides a simple and intuitive solution which is shown to be effective on both synthetic and multiple real-world datasets.
>
> Thank you, this is an exact summarization of the main messages in our paper.
>
> The only thing that is important to emphasize is that our solution is *intentionally the simplest solution*, to demonstrate that over-squashing in GNNs is *so prevalent and untreated* that *even the simplest solution helps*.
> The main point here is the evidence that the *bottleneck exists* in state-of-the-art models, rather than the introduction of the simple solution itself.
> This simple solution opens the path for better solutions for over-squashing and for a variety of research directions.
>
>
> > it is will be still be easier to follow if citations are provided when baseline approaches first appear, e.g., mentioning GGNN (Li et al 2016) and GAT (Velickovic et al 2018) at Page 4.
>
> Thank you, we will fix this in the next revision.
>
> >In Section 4.1, in GAT soft-attention is used, thus the model adaptively combines information from adjacent edges, rather than only consider half of them (which may require hard-attention).
>
> Thank you, we will clarify this in the next revision.

---

### Official Review · AnonReviewer5 · 2020-11-05
**Interesting observation but questionable motivation and too simple solution**

**Rating:** 4
**Confidence:** 4

**Review:**

Overall, the paper identifies a well-known problem in GNN, that is how to incorporate long-range information into GNN computation. The papers propose that this is due to the "over-squashing problem", and provides some arguments and evaluation on this problem. However, I think the motivation is questionable and the solution is too simple.

[Questionable motivation]
In Section 3, the paper starts the motivation as described in Figure 2. The paper argues that "Since the model must propagate information from all green nodes before predicting the label, a bottleneck at the target node is inevitable". However, this example seems trivial for a GNN to solve, and the argument does not make sense to me.
In Figure 2, by simply counting how many blue nodes are within a node's neighborhood, a one-layer GNN can trivially make a correct prediction. Consequently, the whole motivation of Section 3 sounds questionable to me. I think this example should be modified, or at least, explained in a clearer way.
The paper further utilizes NEIGHBORSMATCH as the synthetic benchmark. Unfortunately, this evaluation is not convincing to me since a simple GNN can work well based on my understanding above.

[Too simple solution]
It is unclear to me why "Adding a fully-adjacent layer (FA)" is a good solution to the long-range problem. FA only changes the K-th layer without changing 1 ... K-1 layers. If the over-squashing problem (proposed in this paper) exist, then the input to FA should already severely suffered from the over-squashing problem as there are already K-1 layers computed. This way, it is unclear how FA can get rid of the problem (the paper claims that it can "prevent over-squashing"), as much of the useful information has already been "squashed" and cannot be recovered.
This is not to mention that FA is a very simple trick. To me the technical contribution is too small. I encourage the authors to proposing a more satisfying solution to the proposed over-squashing problem.

[Omitted discussion]
Finally, the paper fails to stress a straight-forward, widely used solution to the long-range problem: adding skip connections in GNN. Such a technique has been widely used in many GNN papers.

---

> ### Author Response · Authors · 2020-11-12
> **Response to AnonReviewer5**
>
> Thank you for taking the time to review our paper!
> You raise important points that we think are addressable within the discussion phase.
> Please see our detailed response below.
>
> > the paper identifies a well-known problem in GNN...
>
> To the best of our knowledge, although the problem of passing long-range messages in GNNs is known, the over-squashing problem itself is **not** well-known.
>
> Although some papers avoided over-squashing using ad hoc solutions (see Section 6),  none of these works explicitly identified the bottleneck and its negative cross-domain implications. Specifically, we show in Section 4.1 that over-squashing prevents the GNN from fitting long-range information starting from a depth as low as 4.
>
> >why "Adding a fully-adjacent layer (FA)" is a good solution to the long-range problem... as much of the useful information has already been "squashed" and cannot be recovered
>
> Adding an FA layer allows both learning the graph topology using the first K-1 layers, and then sending these topology-aware representations to long-range nodes directly.
> Adding an FA layer relieves the first K-1 layers from making an effort to pass long-range messages. The model will not have to optimize these K-1 layers for the objective of compressing long-range messages, because this will be achieved by the last FA layer, which allows a direct passing of long-range messages.
> Thus, useful information can be obtained directly from the original source node, without being squashed.
>
> >the solution is too simple... not to mention that FA is a very simple trick... Such a technique has been widely used in many GNN papers... I encourage the authors to proposing a more satisfying solution to the proposed over-squashing problem.
>
> The solution is **intentionally simple**.
>
> We use the fully-adjacent (FA) layer only to demonstrate that over-squashing in GNNs is *so prevalent and untreated* that *even the simplest solution helps*.
> Thus, showing that this simplest solution *improves the results on extensively-tuned models* is evidence that the original models suffer from over-squashing.
> The important point here is the evidence that the **bottleneck exists** in state-of-the-art models, rather than the simple solution itself.
> This simple solution opens the path for a variety of exciting follow-up improvements and even better solutions for over-squashing.
>
> >In Figure 2, by simply counting how many blue nodes are within a node's neighborhood, a one-layer GNN can trivially make a correct prediction... a simple GNN can work well based on my understanding above.
>
> We may have explained it poorly, but a correct prediction **cannot** be made using a one-layer GNN.
>
> Counting of how many blue neighbors are within a node's neighborhood can be performed in the first GNN layer, but the problem is designed such that this information (the number of neighbors) needs to flow from **all** green nodes **to the target node** to make a correct prediction. The information in the target node must be matched with the information flowing from the other green nodes to make a correct prediction.
> Figure 2 shows the general case, and Figure 5 (in the Appendix) shows the actual case that we trained the GNNs on.
>
> So, the minimal number of GNN layers is as the tree depth (see Figure 5), because otherwise, the model will suffer from "under-reaching" (as we explain in Section 4.2 - the required information will not *reach* the target node). The problem here is that as the number of layers increases, the amount of information that needs to be preserved can grow exponentially.
>
> Practically, as we show in Section 4.1, starting from depth=4, GNNs cannot even **train** to 100% accuracy, and even after tuning various optimization hyperparameters such as changing activations (ReLu, tanh, MLP, none), layer/batch norm, residual connections, batch size, and whether or not the same GNN weights should be "unrolled" over time steps.
> Our code is provided in the supplementary material.

---

> > ### Comment · AnonReviewer5 · 2020-11-12
> > **Response to Authors**
> >
> > "To the best of our knowledge, the over-squashing is not well-known."
> > The authors have misunderstood my review. To me, the underlying targeted problem in this paper is how to incorporate long-range information into GNN computation, which is well-known. It can date back to Jumping Knowledge Networks (https://arxiv.org/abs/1806.03536). Over-squashing is just an explanation provided in this paper -- I don't find solid evidence that this phenomenon does exist.
> > For example, Figure 3 in this paper only shows that the GNN performance degrades, as the tree depth gets deeper. To me, this can only show that GNN fails to incorporate long-range dependency. Over-squashing may or may not be the explanation. I suggest the author provides concrete evidence to show that *it is over-squashing that causes this performance degradation, not other reasons*.
> >
> > "We may have explained it poorly, but a correct prediction cannot be made using a one-layer GNN."
> > I re-read the definition of NEIGHBORSMATCH problem in the manuscript. It said: "The goal is to predict a label for the target node, which is marked with a question mark ( ? ), according to its number of blue neighbors." Following this definition, I think a one-layer GNN can do the job. Indeed, the authors wrote that "information (the number of neighbors) needs to flow from all green nodes to the target node to make a correct prediction", but I would regard this as a possible design by the author. To solve the problem, the model does not need to be that complex.
> > The reason I challenge this setting is that it make the evaluation not convincing. If a one-layer GNN can do pretty well, comparing with deeper GNN does not make sense.
> > If the setting is not what I understood, then at least Section 3 and Figure 2 should be rewritten. The current formulation makes people think that an obvious solution exists.
> >
> > Finally, the authors do not respond to my last concern: "adding skip connections in GNN. Such a technique has been widely used in many GNN papers."
> > I think adding this discussion is very important. Adding a direct experiment will help explain this point. Given a GNN with residual connection, will FA still help?

---

> > > ### Author Response · Authors · 2020-11-13
> > > **Authors' Response**
> > >
> > > Thank you for your quick response. We are happy to see that you have read our paper thoroughly, and we appreciate you investing time in thinking about it.
> > >
> > > The main point of the paper is highlighting a problem in training GNNs - a bottleneck that causes over-squashing.
> > > The paper is about *the problem*. This problem explains some phenomena that were indeed observed in GNNs for years.
> > >
> > > Please see our response below.
> > >
> > >
> > > >To me, the underlying targeted problem in this paper is how to incorporate long-range information into GNN computation
> > >
> > > "How to incorporate long-range information" is **not** the targeted problem in our paper.
> > > The focus of this paper is on highlighting **the problem of over-squashing** and showing evidence for its negative effect, in both synthetic and state-of-the-art models.
> > > We do *not* attempt to provide any "how-to" solution other than the simple FA solution, which is only designed to highlight the prevalence and severity of **the problem**. The problem is so untreated - that even the simplest solution helps!
> > >
> > > >widely used solution to the long-range problem ...
> > >
> > > Calling this problem "the long-range problem" exactly demonstrates the importance of our work.
> > > The purpose of our work is to **identify** this problem, give it a name, explain it, demonstrate the problem's prevalence in SOTA models, measure the problem's effect in a variety of domains, and demonstrate and explain the difference in its effect across different GNN types.
> > >
> > > We believe that just calling this "the long-range problem" and focusing only on solutions is *not* satisfactory, as we as a research community must **understand the problem's cause, and not only its symptoms**.
> > >
> > > > Figure 3 in this paper only shows that the GNN performance degrades, as the tree depth gets deeper. To me, this can only show that GNN fails to incorporate long-range dependency...
> > >
> > > To rule out the possibility that the "GNN fails to incorporate long-range dependency" and that the range itself is the problem, we repeated the experiment of Figure 3 for depths 4 to 8, where the distance remained the same (4 to 8), but the amount of information was as in depth=3.
> > > In other words, the graph looks like a tree of depth=3, where the root is connected to a "chain" of length of up to 5, and the target node is at the other side of the "chain".
> > > This setting maintains the long-range as in the original problem, but reduces the amount of information that needs to be squashed. In other words, this setting *disentangles* of the effect of the long-range itself from the effect of the growing amount of information (i.e., from over-squashing).
> > >
> > > In this setting, **all GNN types could easily fit the data to close to 100%** across all distances, showing that the problem is not the long-range itself, but rather, the growing amount of information (i.e., over-squashing).
> > >
> > > We will include these experiments in the paper.
> > >
> > >
> > > >I think a one-layer GNN can do the job
> > >
> > > Again, we may have explained it poorly. A one-layer GNN **cannot** do the job.
> > >
> > > To solve the problem, the model needs to count the number of blue neighbors at the target node and each of the green nodes, and then predict the correct label by **matching** the number of neighbors of the target node with the green node that has **the same number of blue neighbors**.
> > > This matching can only be performed by propagating the information from all green nodes all the way to the target node.
> > > In every example in the dataset - every green node has a different number of blue neighbors, and the mapping between the "numbers of green nodes" to "alphabetical labels" is **different in every example**. Thus, this long-range propagation and matching *must* be performed for every example.
> > >
> > > Using a single-layer GNN will be able to count neighbors, but will not be able to perform the example-specific matching between the number of neighbors at the target node and the green node that has the same number of blue neighbors.
> > >
> > > If you mean that we could have theoretically changed the graph such that the target node will be directly connected to its matching green node - it would be possible only in this synthetic benchmark,
> > > but in real-life datasets, the whole point is the fact that the required message-passing connectivity is not known apriori, and practically, we can only use the apparent structure (i.e., the molecule's structure).
> > >
> > > We will clarify this in Section 3 and Figure 2.
> > >
> > >
> > >
> > > > Given a GNN with residual connection, will FA still help?
> > >
> > > All experiments in the paper *already include* residual connections, summing every node with its own representation in the previous layer.
> > > An FA layer is orthogonal to residual connections, as the FA layer connects *different* nodes after the same layer.

---

### Author Response · Authors · 2020-11-13
**Changes since the original submission 11/13/2020**

We thank the reviewers for their thorough reviews and useful comments! We feel that they have helped us improve the paper.

We updated our submission and included the following main changes:

1. Fixed presentation comments by AnonReviewer4.

2. Added discussion about over-smoothing and over-squashing, following comments from AnonReviewer3.

3. Improved the caption of Figure 2, as suggested by AnonReviewer3.

4. Included updated results for VarMisuse (Table 3) where we obtain a new SOTA accuracy of 83.8% on the "UnseenProj Test" set. For these new results, the FA layer is the same type as the first layers.
In this setting, all GNNs benefit from the FA layer in the "seen projects" and most GNNs benefit from the FA layer in the "unseen projects".

5. Following the question of AnonReviewer1, we included an ablation where the FA layer is the *penultimate* layer, followed by a standard GNN layer, in Table 5 in Appendix B. This leads to results that are slightly better than a final FA layer.

6. Following the question of AnonReviewer1, we included an ablation that only samples {25%, 50%, 75%} of the edges in the FA layer (where 0% is the "base" model, and 100% is the "+FA" model), in Table 6 in Appendix B. These results show that the full FA layer achieves the largest reduction in error, but even a fraction of the edges improves the results over the base model. Overall, the percentage of used edges in the partial-FA layer is correlated with its reduction in error.

7. Following the suggestion of AnonReviewer5, to rule out the possibility that the long-range itself is the problem, we repeated the experiment of Figure 3 (Tree-NeighborsMatch) for depths 4 to 8, where the distance remained the same (4 to 8), but the amount of information was as in depth=3.
This setting *disentangles* of the effect of the long-range itself from the effect of the growing amount of information (i.e., from over-squashing).

In this setting, **all GNN types could easily fit the data to close to 100%** across all distances, showing that the problem is not the long-range itself, but rather, the growing amount of information (i.e., over-squashing).

8. Following the important suggestion of AnonReviewer5, we clarified (in Figure 2 and Section 3) why a single-layer GNN cannot solve the NeighborsMatch problem. The reason is that in every example in the dataset - every green node has a different number of blue neighbors, and every example has a *different mapping* between the "numbers of green nodes" to "alphabetical labels". Thus, this long-range propagation and matching *must* be performed for every example, and requires at least $depth$ layers.

---

### Author Response · Authors · 2020-11-22
**Any questions before the end of the discussion period?**

We would like to thank again all reviewers, and AnonReviewer5 who engaged in a discussion.

Please let us know if there are additional questions or concerns before the end of the discussion period.

We would be happy to do any follow-up discussion or address any additional comments.

---

### Decision · Program_Chairs · 2021-01-07
**Final Decision**

**Decision:**

Accept (Poster)

**Comment:**

The paper identifies the phenomenon of oversquashing in GNNs and relate it to bottleneck. While this phenomenon has been previously observed, the analysis is new and insightful. The authors conclude that standard message passing may be inefficient in cases where the graphs exhibit an exponentially growing number of neighbors and long-range dependencies, and propose a solution in the form of a fully-adjacent layer. While the paper does not offer much methodologically, it is the observation of bottleneck that is of importance.

We therefore believe that the criticism raised by some reviewers of the observation not being novel and the solution "too simple" rather unsubstantiated. The authors have well addressed these issues in their rebuttal. The AC recommends accepting the paper.